# Archaeometric Approaches to Defining Sustainable Governance: Wari Brewing Traditions and the Building of Political Relationships in Ancient Peru

**Patrick Ryan Williams** [1,*], **Donna J. Nash** [2], **Joshua M. Henkin** [3] **and Ruth Ann Armitage** [4]

1   Field Museum, Chicago, IL 60605, USA
2   Department of Anthropology, University of North Carolina at Greensboro, Greensboro, NC 27412, USA; djnash@uncg.edu
3   Medicinal Chemistry and Pharmacognosy, University of Illinois at Chicago, Chicago, IL 60607, USA; henkinj@gmail.com
4   Department of Chemistry, Eastern Michigan University, Ypsilanti, MI 48197, USA; rarmitage@emich.edu
*   Correspondence: rwilliams@fieldmuseum.org; Tel.: +1-312-665-7008

**Abstract:** Utilizing archaeometric methods, we evaluate the nature of production of feasting events in the ancient Wari state (600–1000 CE). Specifically, we focus on the fabrication of ceramic serving and brewing wares for the alcoholic beverage *chicha de molle*. We examine the source materials used in the creation of these vessels with elemental analysis techniques (INAA and LA-ICP-MS). We then assess the chemical traces of the residues present in the ceramic pores of the vessels to detect compounds indicative of the plants used in *chicha* production (DART-MS). While previous research has identified circumstantial evidence for the use of *Schinus molle* in the production process, this research presents direct evidence of its existence in the pores of the ceramic vessels. We also assess what this material evidence suggests about the sustainability of the feasting events as a mode of political interaction in the Wari sphere. Our evaluation indicates that regional resource use in the production of the ceramic vessels promoted locally sustainable raw material procurement for the making of the festivities. Likewise, drought resistant crops became the key ingredients in the beverages produced and provided a resilient harvest for *chicha* production that was adopted by successor groups.

**Keywords:** elemental analysis; archaeological chemistry; organic residue analysis; phytochemistry; ethnobotany; Andean Middle Horizon

## 1. Introduction

Sustainable governance requires a shared set of values by which political elites affirm their allegiance to a set of ideals. They may differ in their interpretation of how they arrive at those ideals, but in successful states they ultimately cooperate for their political futures [1]. That shared set of values requires an understanding of their common past and of their collaborative potential and those understandings are forged in political discourse. In ancient Peru, and in many other societies, those shared interests were built in ritual acts consummated in special accords.

In the Wari empire (600–1000 CE), those agreements were formed, in part, during elite festivals, involving the consumption of *chicha de molle*, an alcoholic beverage of superb potency. Served in ceramic vessels that invoked the supernatural and or communicated elite allegiances [2,3], *chicha* consumed during ritual drinking sessions fomented political alliances and reified Wari ideology [4]. The most elaborately decorated ceramic cups may have established a link between vessels and solidified the relationships between the Wari elite who drank from them and the supernatural beings that controlled

water availability and fecundity that were presumably represented on these cups. The practice of drinking *chicha* invoked the flow of mountain water and represented the shared desire to exert supernatural control over the most precious resource—water [5].

These rites of incorporation relied on both specialized knowledge on how to produce them, as well as the material means to enact them at the various venues throughout the empire. Specialized knowledge involved the harvesting of clays to build the ceramic bodies and the acquisition of pigments to create the highly ritualized iconography painted on the vessels. It required the esoteric knowledge of the geometric designs and the graphic and vivid supernatural beings painted on the vessels and the means to execute them [6]. It also required a knowledge of the art of brewing and an understanding of the ethnobotanical materials that were the ingredients in a successful beer [7].

The material means to produce these events, in the Wari case, revolved around highly decorated and ritually charged iconography on ceramic brewing and serving wares [8]. Use of renewable local resources may have ensured that the various regional venues for these events were not dependent on resources beyond the regional governor's immediate control. That allowed for these festive events to be independent of disruption to trade routes, political bickering outside the local area, or interference from external adversaries. In other cases, ornate imported ceramic vessels might signal stronger affinity with the imperial center. However, their replenishment would be dependent on external producers distant from the brewing and feasting locales. Interference in the delivery of these goods could impact the ability to carry out the festivals that reaffirmed political ties and alliances.

The other important component for producing a Wari festival was the raw material for the brew itself. *Chicha* can be made of many different ingredients [9]. The most common base for *chicha* in the late prehispanic period was maize, and it remains to this day the most popular ingredient in the Andes. Other *chichas* from distinct areas of the Andes were reported by Spanish chroniclers to be made of tubers, peanuts, strawberries (*Fragaria chiloensis*) and other fruits, quinoa, and the berry of the Peruvian pepper tree, *Schinus molle*. This latter ingredient is especially interesting since finds of large quantities of desiccated *molle* seeds are often found in Wari sites, and provide a compelling indication of the type of *chicha* favored by the Wari [10]. We argue that both the ceramic vessels from which Wari beer was served, as well as the composition of the beer itself, were critical to making these events uniquely Wari. And it was only in these special circumstances that political allegiances sanctioned by Wari customs could be formed.

These accords were materialized both in the media from which the sacred beverage was consumed and in the liquid itself. The ceramic vessels in which the *chicha* was served were especially created for the elite event, and the brew which cemented the agreements conceived was also an extraordinary concoction focused on Wari culinary traditions rooted in local-imperial relations created for the occasion. Likewise, the events themselves were held in Wari architectural frames implanted within administrative centers throughout the imperial realm. These architectural complexes included features like platforms that highlighted the role of the patron of the feast in asymmetrical power relations with the attendees [11].

In this contribution, we utilize archaeometric methods to elucidate the technologies for producing and serving the political elixir for cementing relationships between elites in the Wari realm: *chicha*. In particular, we evaluate the special raw material sources used in the production of ritually decorated ceramics in which the brew was provided, as well as the essential constituents of the brew itself as preserved in the pores of the ceramics from which it was made and served.

## 1.1. Cerro Baúl Brewery as Study Focus

The materials on which this research is based were recovered from an ancient Wari brewery discovered at the site of Cerro Baúl [10,12]. Cerro Baúl was the southernmost administrative center in the Wari realm. Located in the Moquegua region of southern Peru, it occupied the summit of a unique mountaintop locale on the frontier with Wari's imperial rivals, the colonies of the Tiwanaku polity [13,14] (Figure 1). It was thus both a cosmopolitan political center and an embassy to a rival

polity. The nature of political interaction on this frontier was a critical example of Wari state practice, and at its heart was a large-scale brewing facility dedicated to producing a unique Wari brew served in highly decorated drinking vessels made on site [6]. Cerro Baúl provides evidence that Wari colonial governance in Moquegua was maintained in part through a ritual feasting center around *chicha* production. This pattern of governance persisted for four centuries on the southern Wari frontier, and implies the practice, as well as the governmental system it supported, were remarkably sustainable, including during periods of more arid climate.

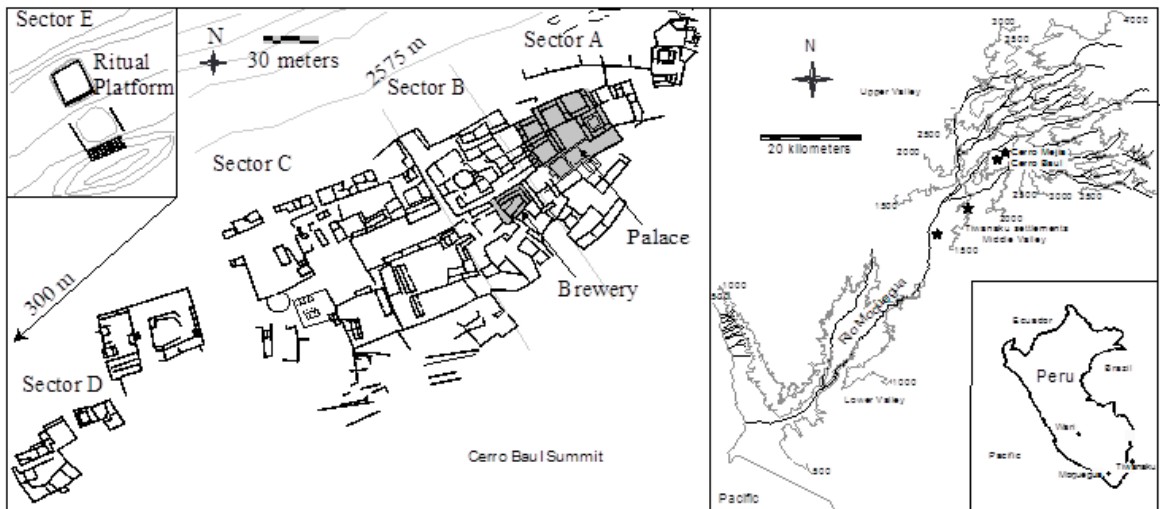

**Figure 1.** Map of the Cerro Baúl site in the Moquegua research area.

Previous research has highlighted the contexts of the brewing traditions and the material remains of production debris [15–17] as well as the nature of Wari political economy [18]. Here we focus on archaeometric analysis of the materials used to create this event, arguing that it was the special contexts of production that made these alliances unique. We assess the nature of the material ingredients that formed the ceramic vessels and the potent brew they contained (Figure 2). This paper uses the results of ceramic sourcing studies and residue analysis of the ceramics from the *chicha* brewery at Cerro Baúl to understand how this practice sustained political interactions over the latter half of the first millennium CE.

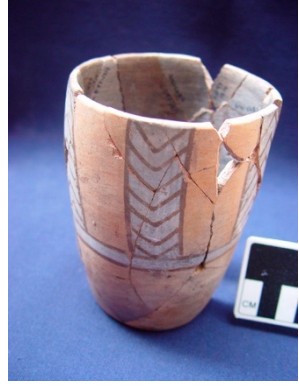
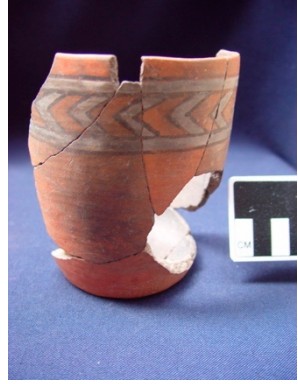

**Figure 2.** *Cont.*

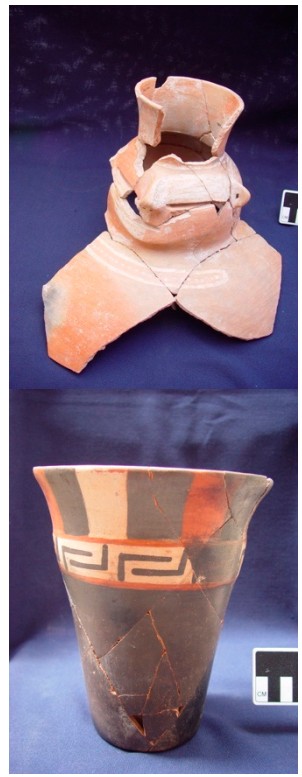
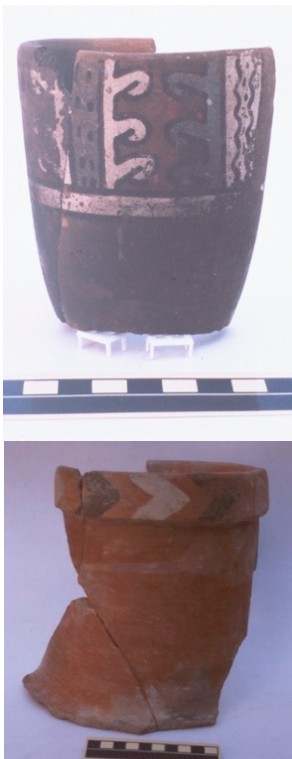

**Figure 2.** Example vessels from the Cerro Baúl brewery identified as part of the Baúl chemical group by INAA. They represent cups, fermentation and serving vats, and a *kero* or drinking mug.

*1.2. The Brewery Context*

The excavated area of the brewery constitutes approximately 500 square meters encompassing four distinct spaces (Rooms 1–4) as well as an additional set of smaller rooms to the southwest (Figure 3). Distinct rooms were dedicated to grinding maize, preparing comestibles, boiling the mash, and fermenting and storing the brew. The boiling room contained the remains of between eight and twelve hearths along the northwest wall, each framed by two upright stones placed 50 cm from the wall to support a ceramic boiling vat. The hearths contained the burnt remains of fuel ash, identified principally as dung, including guinea pig as well as camelid sources. Trash pits in the floor of the boiling room contained the remains of large numbers of the spent drupes of the Peruvian pepper berry, *S. molle* [16]. The fragmented remains of large cooking vessels were also recovered from this space, with sooted base fragments and clear evidence of use. One of these fragments was collected in situ and wrapped in foil for residue analysis, reported below.

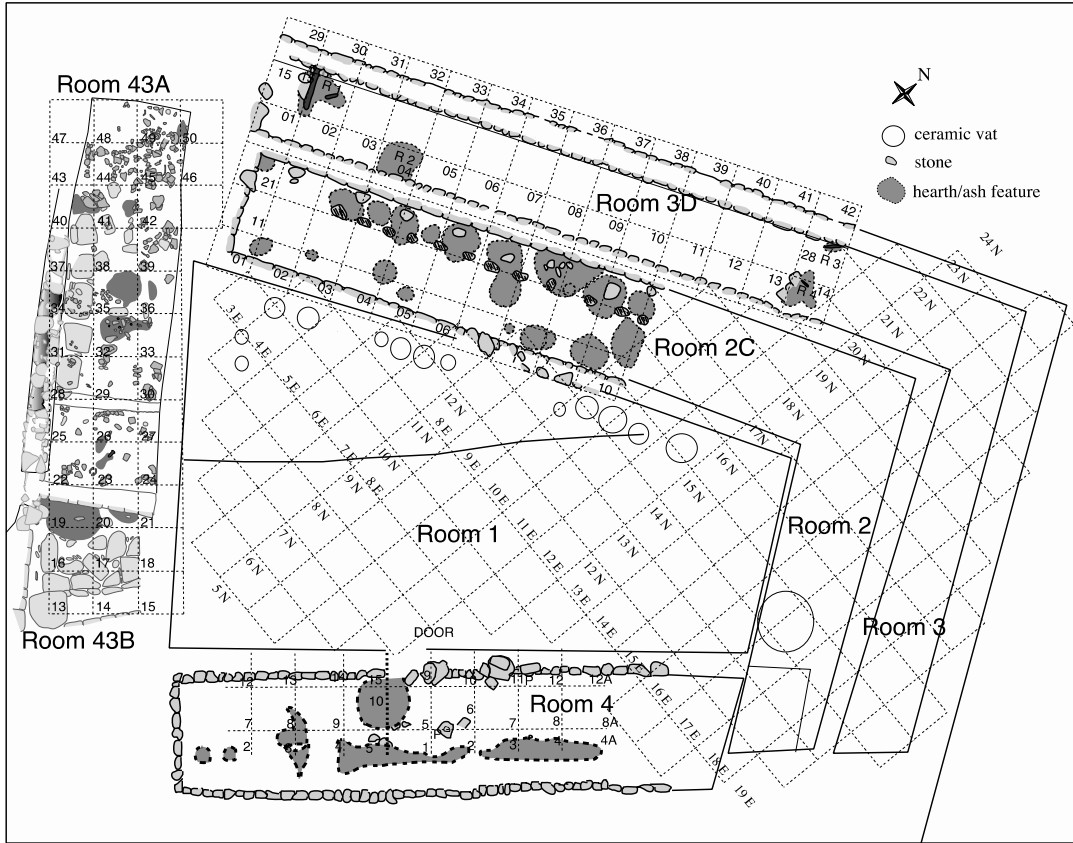

**Figure 3.** Map of the brewery at Cerro Baúl, illustrating the fermentation/storage area (Room 1), boiling area (Room 2C), and the grinding and food preparation rooms (Rooms 4, 43A). Other rooms may have been used for storage or access to brewing facilities.

The adjacent fermentation or storage room contained the remains of several oversize jars, originally lined up along the northwest wall. These jars held up to 150 L of brew each and were decorated as Wari personages, with a face on the neck of the jar, or with a chevron headband painted on the rim of the neck as shown in Figure 2. Another large vat found in situ embedded in the floor was over a meter in diameter and may have held up to 1000 L of liquid. The fermentation room and the adjacent grinding rooms were the areas where most of the fineware fermentation vats and drinking and serving vessels were found smashed on the floors [10]. These serving vessels include cups of ca. 300 mL volume, keros (drinking mugs) of upwards of 2-L volumes, open bowls, and restricted jars. Four of the sherds from the fermentation/storage vessels were exported for residue sampling as described later. The highly decorated nature of the assemblage, the comparative high quality of the ceramic materials, and the highly privileged location of the brewery indicates it was an elite production space. While evidence of smaller scale brewing has been identified in intermediate elite contexts at the adjacent site of Cerro Mejia and Cerro Trapiche and in the houses of Tiwanaku leaders, we have not seen contemporary commoner houses with these production contexts.

Radiocarbon dates and evidence of modification of space in the brewery indicate a long occupation, with at least two phases of construction. Radiocarbon dates range from 1400 ± 60 BP to 900 ± 40 BP uncalibrated [10], which span the range of occupation of the Wari settlement as a whole. When the site was abandoned ca 1050 CE, the brewery was intentionally burnt to the ground and the fineware ceramic drinking vessels were smashed in the smoldering flames. As the fire extinguished, seven semi-precious stone and shell bead necklaces were placed atop the ashes and covered with sediment to cap the ritual closure of the facility. The large fermentation jars in the central storage room were also broken and pieces were dispersed throughout the brewery. These drinking cups, serving wares, and fermentation

vats were the samples for the brewery ceramic paste sourcing analysis discussed next, some of which are displayed in Figure 2.

## 2. Materials and Methods

### 2.1. Ceramic Paste Analysis Methods

A sample of 20 ceramic sherds from the Baúl brewery were analyzed by instrumental neutron activation analysis (INAA) at the Missouri University Research Reactor (MURR) and another 19 samples from nearby consumption contexts were analyzed by both INAA at MURR and by laser ablation inductively coupled plasma mass spectrometry (LA-ICP-MS) at the Elemental Analysis Facility (EAF) at The Field Museum [19–21]. We also analyzed an additional 41 sherds from other summit contexts on Cerro Baúl for a total of 80 ceramic samples from the top of the mountain citadel (Table 1).The INAA dataset also contains comparative ceramic samples of 10 sherds from Formative sites in the local Moquegua region, 51 sherds from the Wari site of Cerro Mejia, 44 samples from Wari and Tiwanaku sites on the slopes of Cerro Baúl, as well as 29 ceramic sherds from the Wari heartland [19].

**Table 1.** Summary table of ceramic sourcing samples from Cerro Baúl and comparative sites.

| Context | N | Lab Numbers | Method | Group(s) Assigned |
|---------|---|-------------|--------|-------------------|
| Baúl brewery (elite) | 20 | PRW002,4–8,11, 13–14, 16, 20, 27, 29–32, 36, 38–40 | INAA | Baúl group (90%)<br>Mejia A (5%)<br>Unassigned (5%) |
| Baúl palace and consumption (elite) | 19 | PRW101–102, 104, 106, 109, 112–113, 118, 120, 123, 125, 131, 133, 136, 144–145, 147–148, 150 | INAA & LA-ICP-MS | Baúl group (100%) |
| Other Baúl summit (elite) | 41 | PRW001, 3, 9–10, 12, 15, 17–19, 21–26, 28, 33–35, 37,103, 105, 107–108, 110–111, 114–117, 119, 121–122, 124, 126, 130, 132, 134, 138, 143, 149 | INAA | Baúl group (46%)<br>Mejia A, B, D, E, or G (29%)<br>Unassigned (25%) |
| Formative sites (local) | 10 | PRW183–192 | INAA | Mejia A (40%)<br>Mejia D (30%)<br>Unassigned (30%) |
| Cerro Mejia (local) | 51 | PRW046–096 | INAA | Mejia A (24%)<br>Mejia B (31%)<br>Mejia D, G, or F (12%)<br>Baúl Group (6%)<br>Unassigned (27%) |
| Baúl slopes (local) | 44 | PRW099–100, 127–129, 135, 137, 139-142, 146, 151–182 | INAA | Mejia A (52%)<br>Mejia B, C, D, E, or F (32%)<br>Baúl Group (2%)<br>Unassigned (14%) |
| Tiwanaku sites in Moquegua | 29 | TW001–029 | LA-ICP-MS | Local Tiwanaku (86%)<br>Unassigned (14%) |
| Wari heartland | 29 | PRW195–223 | INAA & LA-ICP-MS | Wari-1 (55%)<br>Wari-2 (21%)<br>Wari-3 (10%)<br>Mejia A (4%)<br>Unassigned (10%) |

Sharratt et al. [22] conducted a clay survey in 2005, providing 50 clay locales that were analyzed via LA-ICP-MS. In conjunction with this analysis, they analyzed an additional 29 Tiwanaku sherds from the local region and compared them with 19 of the 29 sherds from the Wari heartland to provide the link

between local clay sources and ceramic production workshops [20–22]. This analysis complements the INAA data and the 19 samples from Baúl consumption contexts outside the brewery mentioned earlier.

INAA analytical methods are detailed in Glascock [23], while we describe ICP-MS methods here. A Varian inductively-coupled plasma-mass spectrometer (ICP-MS), equivalent to the Varian 810 instrument, was used at the EAF (quadrupole, auto-optimized spectrometer on 55 selected isotopes). The facility uses a New Wave UP213 (helium carrier gas, 213 nm laser operated at 0.2 mJ and a pulse frequency of 15 Hz) laser in conjunction with the LA-ICP-MS to introduce solid samples. Methodology follows that presented previously [20,22].

Ceramic samples were laser-ablated with a spot size of 150 microns and a dwell time of ninety seconds. Each sample was ablated ten times and a total of 55 elements was measured, using 29Si as an internal standard to control for time variability in ablation efficiency and resulting signal strength. Concentrations were calculated using NIST standards n610, n612, and Brick Clay (n679) via the approach first presented by Gratuze, Blet-Lemarquand, and Barrandon [24]. Large temper grains in the ceramic were avoided when positioning the laser so as to focus on the clays used in production. Ablation took place on sherd cross-sections to avoid pigments and treatments on ceramic surfaces.

Statistical analysis was performed using SPSS and Gauss Runtime statistical routines developed by MURR.A principal components analysis was carried out on the 16 elements that best produced group separation in the INAA and ICP-MS data [20]. We converted concentration values to base-10 logarithms to improve normality for trace element data and to minimize scaling differences of trace elements. We conducted a principal components analysis using elements from the variance–covariance matrix in order to examine the multivariate patterns in the data.

## 2.2. Residue Analysis Methods

In order to provide a baseline for residues within Wari ceramic vessels, Donna Nash conducted an experimental archaeology process for the production of *chicha de molle*. In collaboration with ethnographic informants, Nash prepared three types of *chicha* in newly fabricated clay vessels: *chicha de jora* (*maíz*), *chicha de molle*, and *chicha de maíz y molle*. Maize was germinated, then ground and boiled. *Molle* was winnowed and the seeds steeped in hot water for a short period of time. The liquid was then left to repose in clay fermentation vessels for five days before tasting. The *chicha* was left in the vessels for another five days, its maximum shelf life before consumption, and the empty clay vessels for each type of *chicha* were broken into sherds. Several years later, the clay vessel sherds were subjected to chemical analysis of residues embedded in the ceramic pores, as described herein. Likewise, five fragments from clay vessels from the ancient brewery at Cerro Baúl were subjected to the same residue analysis to ascertain which brew was chemically most similar to the archaeological residues embedded in the ceramic vessels.

Direct analysis in real time mass spectrometry (DART-MS) was employed to characterize potsherd residues and spent *molle* drupes recovered from the Cerro Baúl brewery, which were compared to experimental archaeology controls (produced by Nash and her ethnographic informants as described above).Analyses were carried out using a DART ionization source (IonSense, Saugus, MA) on a JEOL AccuTOF mass spectrometer, additional background on the DART-MS may be found elsewhere [25]. Spectra were collected in negative ion mode with the DART ion source at 500 °C using helium as the DART gas. Orifice 1 on the AccuTOF was set to −30 cV to minimize fragmentation; the ring lens voltage and orifice 2 were set to −5 V. The temperature at orifice 1 was held at 150 °C for analysis. The DART grid voltage was kept at the default value of −530 V. The mass spectrometer RF ion guide ("peaks voltage") was set to 800 V to maximize sensitivity above $m/z$ 80. Mass calibration was carried out using PEG-600 in methanol during each acquisition. The mass resolving power was approximately 6000.

Individual *molle* drupes were held in a pair of tweezers and introduced into the space between the DART ion source and the AccuTOF mass spectrometer orifice. No sample preparation was required. For the ceramic materials, a small amount of the surface was removed using a rotary grinding tool

and placed into a small autosampler vial with a few drops of HPLC grade methanol. This ceramic slurry was introduced into the space between the DART ion source and the mass spectrometer orifice on the closed end of a capillary melting point tube. The methanol served primarily to allow a significant quantity of ceramic slurry to be subjected to ionization rather than as a solvent for the residue [26]. DART-MS has a distinct advantage over the traditional residue analysis methods like gas chromatography-mass spectrometry (GC-MS) and liquid chromatography-mass spectrometry (LC-MS) in that it requires no sample preparation, or as in the case here, very little. The speed of analysis—just a few seconds in the ion source—comes at a cost: molecules can only be identified based on their molecular mass. Fragmentation can be accomplished by using a higher voltage on Orifice 1, but interpretation is complicated by the sheer number of ions produced from the mixture being ionized. This analysis was an initial attempt to determine if the pertinent residues could be distinguished with DART-MS, an approach that uses minimal solvent, no additional reagents, and requires very little time. It is, one could argue, a sustainable initial approach to screening ceramics prior to further study with more traditional methods like GC-MS or LC-MS.

The sensitivity of the DART-MS method is dependent upon the amount of material presented for ionization. For the ethnographic samples, the control ceramic and *chicha de jora* vessels were sampled both before and after being scrubbed with a brush under running water to simulate the process of field cleaning of ceramic samples. Each sample was run in triplicate unless otherwise noted. Compounds were identified in the spectra using modified versions of the databases compiled by one of us (JMH) of compounds that have been previously identified and isolated in *Schinus* spp. and in maize, from Reaxys and/or NAPRALERT web-based tools [27]. The modified databases removed compounds present as glycosides (or linked to other sugars) as these compounds do not readily ionize by DART-MS.

Identifications are based on the observed peaks being within 10 millimass units ($\pm1$ in the second decimal place) of the calculated exact mass of the compounds in the database. Each compound is identified based on only one mass, as the deprotonated species (molecule less one hydrogen atom, or M–H anion); as multiple compounds can share the same formula and thus the same mass, identifications must also be based on what compounds are present from this small subset derived from web-based tools and the analysts' (RAA and JMH) discernment in relation to the mass spectra.

## 3. Results

### 3.1. Ceramic Paste Sourcing Results

We discovered that Wari ceramics from Cerro Baúl were chemically distinct from contemporary materials in the local region, including both Tiwanaku and local sherds.They were also distinct from a control group of ceramics excavated at the Wari capital in Ayacucho. Statistical analysis of the INAA data resulted in the definition of three chemical groups from the Wari heartland (Wari-1, 2, and 3), one chemical group concentrated on the Cerro Baúl site summit (Baúl Ref.), and seven groups of local ceramic production (Mejia A-G) [19].In the INAA data, a biplot on the first two principal components calculated from the variance–covariance matrix of the ceramic samples shows the brewery ceramics can be distinguished from both Wari heartland materials and other local ceramics on an axis of variability associated with the second principal component (Figure 4).

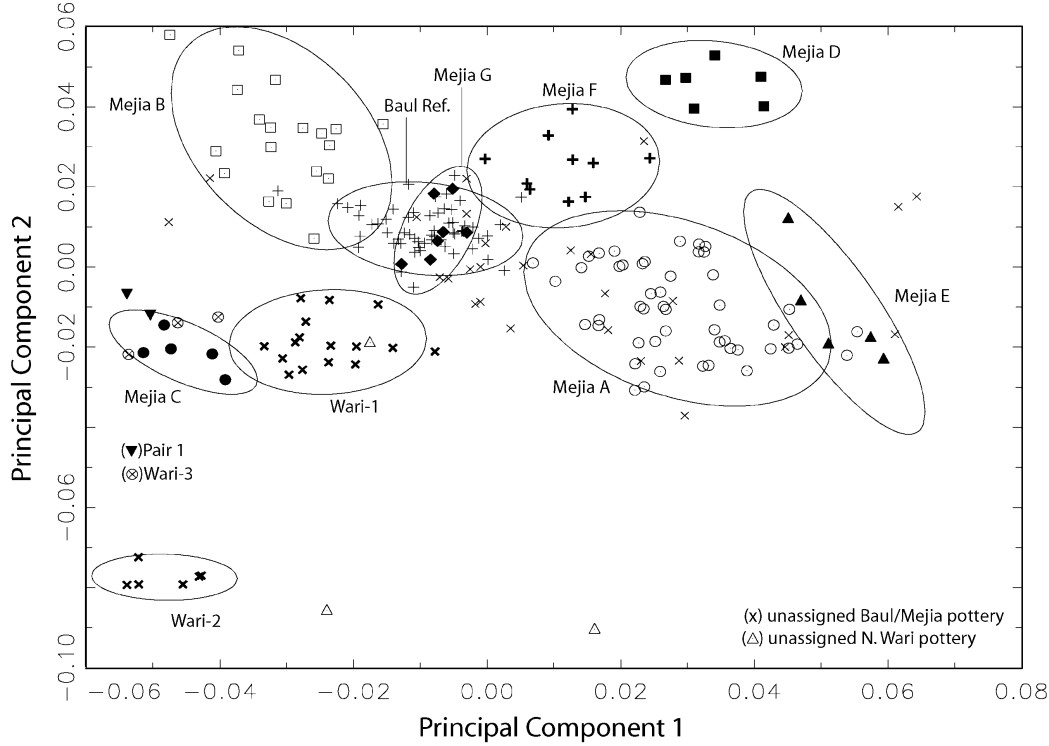

**Figure 4.** PC1 v PC2 for INAA data from Baúl, Mejia (local) and Wari heartland ceramics [19].

This component is positively loaded on Na, Sr, Ca, and Th and negatively loaded on Fe, V, Co, Sc, Ti, Sb, Cr, As, and Zn. A biplot of Cr and La distinguishes the Baúl group from the local Mejia pastes (Figure 5).

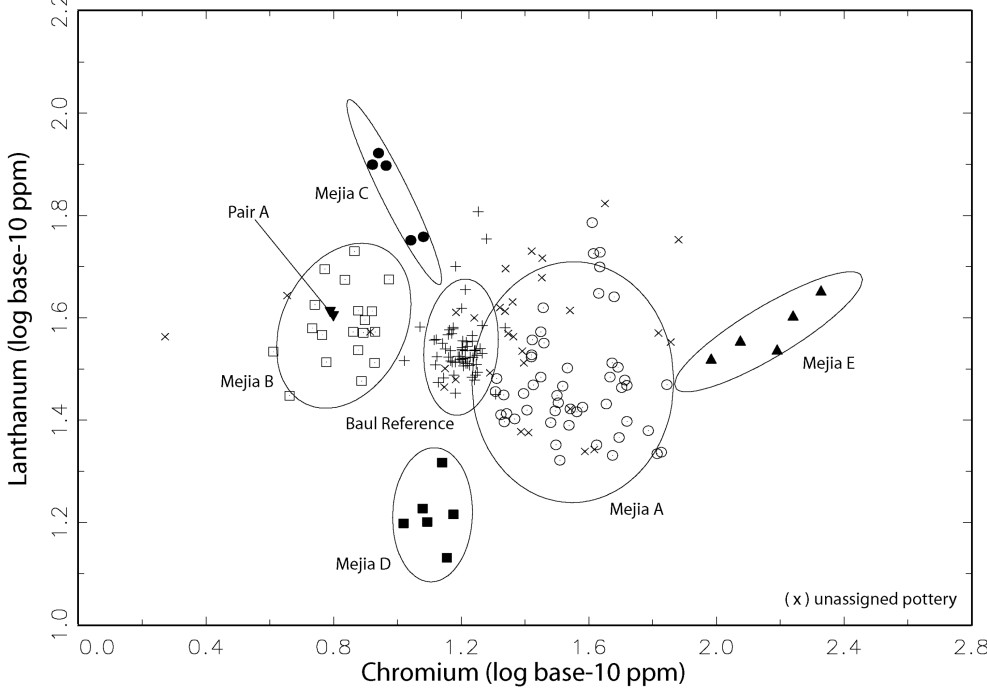

**Figure 5.** Chromium v Lanthanum plot for INAA data [19].

In the ICP-MS data, the distinction of the Cerro Baúl brewery ceramics is also pronounced, but this time the second principal component, which together with PC1 accounts for 47% of the total variance,

is loaded positively on Be, K, Nb, Rb, U, and Th. It is negatively loaded on Fe, Cu, Sr, V, Co, Mg, and Ca [22]. Analysis of local clay sources indicate that the Baúl ceramics match most closely with a clay source across the valley from the site. The local Tiwanaku and Mejia ceramics best match clay sources from geological formations in the Torata and Moquegua valleys themselves [22]. None of the archaeological ceramics from the Moquegua region were chemically similar to the three sources identified from the Wari heartland.

Ceramic samples from the brewery (*n* = 20) represent 25% of the collection of pottery sourced by INAA on the summit of Cerro Baúl (*n* = 80). While the Baúl chemical group accounts for about 70% of the summit assemblage, it accounts for all but two samples from the brewery. Thus, 90% (18/20) of the brewery ceramic vessels are from the Baúl ceramic chemical group (Table 1). Most of those are serving wares constituted by bowls and drinking cups. It is apparent that the brewery was a key consumer of the special ceramic workshop production represented by the Baúl chemical group. The palace and adjacent consumption contexts on the summit of Cerro Baúl near the brewery were also key consumers of the Baúl workshop ceramic wares. Nash [6] identified the probable location of a ceramic workshop in the palace on the summit of the mountain that is the likely locale of production for these specialized ceramic vessels.

Off the summit of Cerro Baúl, the finely made and decorated Baúl group ceramics are very rare [6]. At the adjacent Wari site of Cerro Mejia, only 6% of the samples match the Baúl group, while in local settlements on the slopes of Cerro Baúl, only 2% (a single vessel) of the ceramic assemblage is part of the Baúl group [19]. Outside of these sites immediately adjacent to the Wari administrative center at Cerro Baúl, none of the other Moquegua valley samples match the Baúl chemical group. In fact, greater than 90% of all these ceramic assemblages were produced from Moquegua or Torata valley clay sources with the exception of the elite wares that are part of the Baúl group from the summit of the mountain.

Thus, the archaeometric data indicate that the brewery ceramics were produced from a specialized set of raw materials, dedicated to political consumption events in ritually charged monumental architecture on the summit of Cerro Baúl. We previously identified the likely clay source for that ceramic production on a mountain across the valley from the site [22]. Interestingly, that clay source is uniquely used by the ceramicists at Cerro Baúl and is not used by any other contemporary groups, despite the high quality of the clays. This likely demonstrates an exclusionary control over the access to the clay sources by the Wari elite.

*3.2. Residue Analysis Results*

The results of the residue analysis of the ethnographic ceramics by DART-MS are presented in Table 2. Several compounds from the *Schinus* database were found in a control vessel that contained none of the *chicha* recipes and were thus excluded from further results as being not indicative of the presence of *Schinus molle* residues. The base peak of all of the *chicha de jora* samples (even after washing) is observed at *m/z* 341.108. Of the possible formulas calculated for this mass was $C_{19}H_{17}O_6$, which would correspond to a number of tetrahydroxyflavone compounds, or $C_{12}H_{22}O_{11}$, which would correspond to a disaccharide like maltose. While the calculated masses for each of these compounds is different, it differs only in the third decimal place, meaning that within the mass resolution of the AccuTOF mass spectrometer, we cannot differentiate between these compounds. Maltose is the most likely source of that peak.

The *chicha de molle* experimental samples contained few of the compounds that were both found in the literature for *Schinus* species and could be ionized by DART-MS. Fewer yet were identified in the *chicha* with both corn and *molle*. The potentially relevant compounds that could pertain to *molle* which seem to have been identified in ethnographic controls of (1) *chicha de molle* alone and (2) *chicha de molle* with maize were respectively: (1) terebanene, teredenene, or β-spathulene (a sesquiterpenoid); safrole (a phenylpropanoid); and gallic acid (a phenolic acid) and (2) terebanene, teredenene, or β-spathulene (a sesquiterpenoid). None of these compounds alone or altogether as an assemblage affords a firm identification of *molle* to these ethnographic controls, but all are consistent with *molle*. Indeed, the DART

mass spectra of the *molle* drupes showed all of these compounds, as well as a compound with $m/z$ 453.332, which corresponds to the $[M-H]^-$ ion for the formula $C_{30}H_{46}O_3$. This is most likely either (1) masticadienoic acid (also known as terebinthone), (2) a positional isomer of that compound such as isomasticadienoic acid, or otherwise (3) moronic acid; these compounds are indistinguishable based on the molecular mass. The DART mass spectrum of the drupes from the Cerro Baúl excavation also showed an additional signal most likely arising from β-sitosterol ($m/z$ 413.380, $C_{29}H_{50}O$ $[M-H]^-$).

**Table 2.** Experimental ceramic residues on vessels from DART-MS.

| *Chicha de jora* **(maize)** | | | | |
|---|---|---|---|---|
| **Compound** | **Formula** | **Measured Mass** | **Calculated Mass** | **Intensity** |
| Methyl gallate | $C_8H_8O_5$ | 183.024 | 183.029 | trace |
| Shikimic acid | $C_7H_{10}O_5$ | 173.041 | 173.045 | trace |
| *Chicha de molle* | | | | |
| **Compound** | **Formula** | **Measured Mass** | **Calculated Mass** | **Intensity** |
| Safrole | $C_{10}H_{10}O_2$ | 161.067 | 161.060 | low |
| Gallic acid | $C_7H_6O_5$ | 169.016 | 169.014 | trace |
| Terebanene, teredenene, or β-spathulene | $C_{15}H_{22}$ | 201.156 | 201.164 | trace |
| Palustrol/rosifoliol/shyobunol | $C_{15}H_{26}O$ | 221.192 | 221.191 | low |

Identifications of triterpenoids that are chemotaxonomically characteristic especially of certain anacardiaceous resins (*Pistacia* spp., *Rhus* spp., and *Schinus* spp.) are particularly heartening. In J. Henkin's view these triterpenoids are, where present in the DART-MS spectra, the best phytochemical evidence for *chicha de molle* production from the archaeologically recovered ceramics and *molle* drupes analyzed (Figure 6; Table 3). Hydroxymasticadienoic acid, (iso)masticadienoic acid, moronic acid, schinol, and/or simiarenol together as this relevant triterpenoid group underlie a solid chemotaxonomic argument for *molle* processing and *chicha de molle* production at Cerro Baúl. These triterpenoids should be regarded as *molle* biomarkers in further research along these lines (Figure 7). Sesquiterpenoid and triterpenoid biomarkers identified, plus the presence of the phenylpropanoid safrole, support the archaeological evidence for *chicha de molle* production at Cerro Baúl.

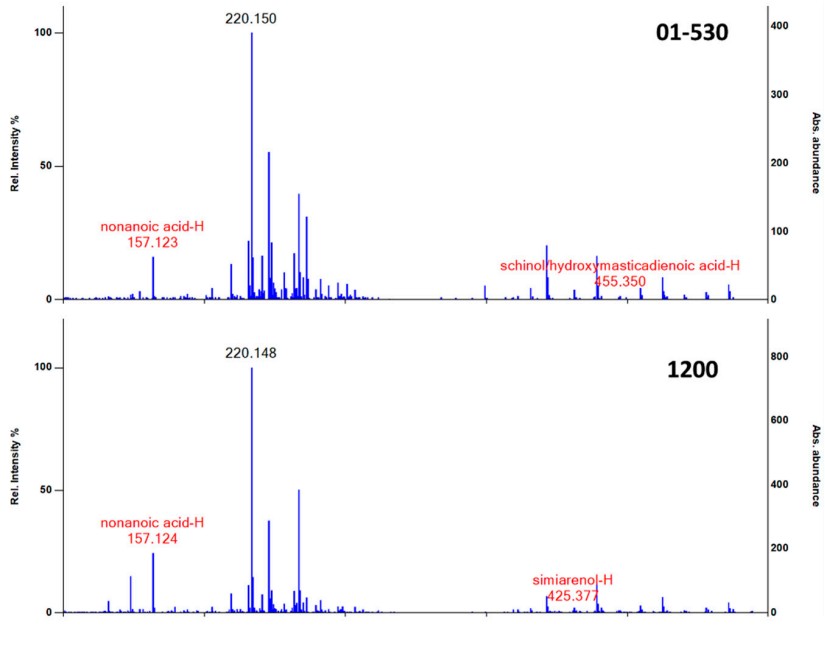

**Figure 6.** *Cont.*

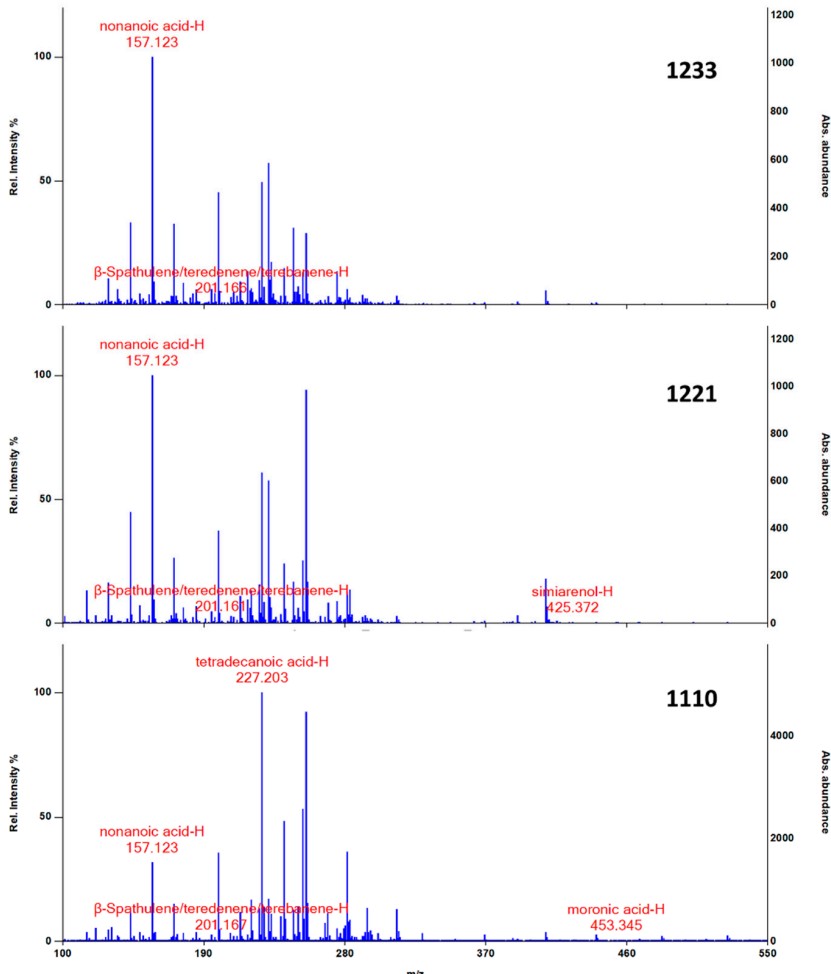

**Figure 6.** Example DART-MS spectra for one replicate each of the Cerro Baúl brewery ceramic residue samples.

**Table 3.** Ceramic residues on archaeological sherds from DART-MS (nd = not detected). Each + indicates a positive identification (based on mass for the [M–H]$^-$ ion) in one replicate. Thus ++ means that the compound was identified in two replicates. For CB89-1221, only two replicate analyses were carried out.

| Compound | Formula | Measured mass, [M−H]$^-$ | Calculated Mass, [M−H]$^-$ | CB04-01-0530 | CB89-1110 | CB89-1200 | CB89-1221 | CB89-1233 |
|---|---|---|---|---|---|---|---|---|
| simiarenol | $C_{30}H_{50}O$ | 425.376 ± 0.002 | 425.378 | nd | nd | + | + | nd |
| masticadienoic acid (terebinthone) or moronic acid | $C_{30}H_{46}O$ | 453.345 ± 0.000 | 453.337 | nd | + | nd | nd | + |
| hydroxymasticadienoic acid | $C_{30}H_{48}O_3$ | 455.348 ± 0.002 | 455.353 | nd | nd | + | nd | nd |
| terebanene, teredenene, or β-spathulene | $C_{15}H_{22}$ | 201.163 ± 0.003 | 201.164 | nd | + | nd | + | ++ |
| safrole | $C_{10}H_{10}O_2$ | 161.061 ± 0.009 | 161.060 | nd | nd | nd | + | + |

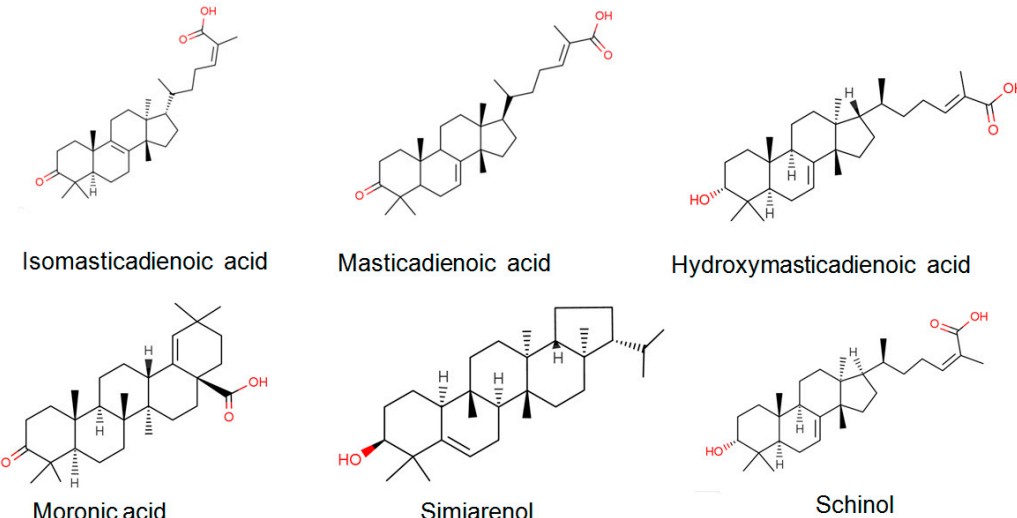

**Figure 7.** Triterpenoid compounds identified by DART-MS in *molle* samples, diagnostic especially for *Schinus* spp. and its close relatives [26].

The results in Table 3 show only the compounds that, based on literature sources, are indicative of *Schinus molle* and that are not found in the control samples that contained only corn-based *chicha*. In none of the spectra were the identified *Schinus* compounds the most intense signal, referred to as the base peak of the spectrum (Figure 6).For two of the sherds the observed base peak was at $m/z$ 220.150 (CB04-01-0530 and CB89-1200), a compound that we were unable to identify, but appears to give primary fragments at $m/z$148 and 205 when Orifice 1 on the AccuTOF is set to $-90$V.The intensity of this signal may indicate it is a contaminant from the environment or handling. For the remaining three sherds, the base peak was most likely a fatty acid. These include myristic acid (tetradecanoic acid, $m/z$ 227.201) in replicates 1 and 2 of CB89-1110, palmitic acid (hexadecanoic acid, $m/z$ 255.232) in the third replicate of both CB89-1110 and -1233, and the shorter-chain pelargonic acid (nonanoic acid, $m/z$ 157.123) in both replicates of CB89-1221 and the second of -1223. None of these fatty acids are observed in significant quantities (based on signal intensity) in the control vessels.

Interestingly, the medium-chain fatty acids are absent in the ceramic vessel that contained the ethnographic *chicha de jora* but dominate the spectra of the ones containing *chicha de molle*, even when corn was also present. Caproic, caprylic, and pelargonic acids (hexanoic, octanoic, and nonanoic) are the top three intensity peaks in the DART-MS spectra of all of vessels that contained the ethnographic *chicha* made with *Schinus molle* berries. Medium-chain fatty acids, while not anticipated by the initial chemoinformatic search that has driven the residue analysis aspect of this research into the material culture of Cerro Baúl, do appear prominently in ethnographic and archaeological *chicha de molle* DART-MS spectra. Truly accounting for the fatty acids' prominence in terms of the phytochemistry of *S. molle* itself, the beverage processing, and/or other environmental inputs would be a sensible focus for any follow-up research. These fatty acids, specifically their isotopic composition, are an excellent target for further analyses. Confirmation of the exact identities of the terpenoids and phenolics would further increase confidence in the identification of the *molle* residues.

## 4. Discussion

How do unique or rare materials contribute to sustainable governance? In reality, it is the know-how in this case that creates the unique experience, not the material itself. Clay sources utilized to produce brewing vessels may be a limited resource or may be distant or difficult to procure. But in the case of Cerro Baúl, the clay source identified by the chemical data was abundant and located less than half a day's walk from the brewery. Wari ceramic specialists did establish a privileged access to this source and created a recipe for working the material. Yet, they did not overexploit the source nor did they focus on incorporating materials that were not readily available in the local area.

The clay itself was a local resource, and it was particularly reserved by the Wari for their specialized workshop in the region. Wari ceramic specialists did not rely on extra-local materials (except perhaps for some pigments) for their ceramic production, nor did they rely on imported ceramic vessels with their ritually charged iconography. They decorated these vessels themselves with highly significant ritual iconography, representing Wari supernatural beings and geometric motifs. The means of forming and decorating these vessels replicated ways from the Wari heartland and did require specialized knowledge. That knowledge was reproduced locally to create the special vessels for the feasting events. Having a local source for the ceramics made the Baúl brewery a sustainable producer of both the containers for production and consumption of the beer itself since they were not reliant on foreign materials.

Yet, these ornate, regionally produced ceramic vessels did signal strong affinity with the imperial center. They were not dependent on external producers distant from the brewing and feasting locales in order to provide the vessels for these political events. They were not subject to disruptions in long distance supply chains, external political interference, nor disasters in distant production locales for these important constituents of the Wari feast. They were, however, dependent on the Wari specialist potters who knew how to create the Wari vessel forms and to decorate them with the highly stylized ritual iconography. These specialists had to experiment with local resources: tempers, clays, and pigments, in order to transform those raw materials into legitimate and authentic Wari ceramic wares. This local production of highly specialized containers provided a high degree of locally sustainable fabrication of genuine Wari ritual receptacles. That was critical to the reproduction of the Wari political economy, not only at Cerro Baúl, but likely in every major Wari center throughout the empire.

The other material components of crucial importance to the brewery's production were of course the grains and fruits fermented to make the political beverage, selected by the Wari specialist brewers for their recipes. Corn is a fairly water intensive crop to produce, though yields are quite high. Other native grains like quinoa have lower yields (500 kg per acre), but may be less water intensive. In modern farming, corn generally requires more water per field than many other native Andean crops, including other grains, tree fruits, potatoes, and other potential *chicha* producing ingredients. Modern corn, though, does provide high calories per unit area (up to 15 million calories per acre) and thus produces significantly more energy despite its higher water requirements.

The pepper berries are a renewable resource as well, available on trees that naturally form hedgerows and are commonly available on the edges of cultivated fields. Growing to heights up to 15 m, they are also hearty and resistant to drought conditions. In an environment of climatic variability, they are a raw material resource that is resilient in its availability. While the fruits are especially prevalent after extended periods of heavy rain, they are present on trees in the study area year round [9]. Thus, while maize is a fairly water intensive crop, *molle* is a drought resistant tree that consistently produces fruit. In addition to the cultural properties *molle* contributes, it also provides a consistent ingredient for *chicha* that is drought tolerant.

The Middle Horizon (600–1000 CE) was generally a period of climatic stability with some variability in precipitation pronounced in later times. Wari terracing technology provided some mitigation of drought conditions through more efficient water use [28]. Towards the end of the period, increased variability in precipitation lead to localized conditions of water deficiency, which in the succeeding centuries became severe and prolonged droughts [29]. The use of *molle* as a *chicha* product seems to be pronounced in the final years of the Baúl brewery's production, slightly after 1000 CE, evidenced by the fact that the residue results presented here come from ceramics used in the ceremonial closure of the brewery and that *molle* dumps are associated with the final use of the floor in the brewery. This may reflect the reliability of *molle* as a resource, even in times of drought stress.

Thus, it was both local resourcing of ceramic vessels for brewing and serving the alcoholic beverage and a focus on a set of brewing ingredients that were locally produced and resilient to climatic variations in production that created a sustainable brewing operation for four centuries.

These variables were incredibly important to the Wari political economy and to the building of local allegiances that sustained imperial relationships over decades. Archaeometric data confirms that local source material was utilized in both ceramic production and beverage fabrication. Yet, it was the ceramicists' specialized knowledge on how to construct and decorate the elaborate drinking vessels that made these events quintessentially Wari. It was also the brewers' knowledge of the culinary recipes of different *chicha* production strategies across dispersed areas of the Andes mountains that brought that shared political identity to strangers in many lands. Local resourcing joined with shared knowledge over vast areas to create the political unity that Wari represented as the Andes' first empire.

**Author Contributions:** Conceptualization, P.R.W. and D.J.N.; Formal analysis, P.R.W., D.J.N., J.M.H., and R.A.A.; Funding acquisition, P.R.W., D.N., and R.A.A.; Investigation, P.R.W., D.J.N., J.M.H., and R.A.A.; Methodology, P.R.W., J.M.H., and R.A.A.; Writing—original draft, P.R.W., D.J.N., and J.M.H.; Writing—review & editing, J.M.H. and R.A.A.

**Funding:** Field research on the Cerro Baúl brewery was funded by the National Science Foundation (BCS-0074410, BCS-0226791), the National Endowment for the Humanities (RZ-50098), the G. A. Bruno Foundation, and the Grainger Foundation. DART-MS analyses were carried out on EMU's AccuTOF mass spectrometer, funded under NSF MRI-R$^2$ (0959621).LA-ICP-MS analyses were carried out on TFM's Varian inductively-coupled plasma-mass spectrometer funded under NSF MRI (0320903).

**Acknowledgments:** INAA was conducted at MURR under the direction of Michael Glascock and preliminary interpretation of results was done by Jeff Speakman. LA-ICP-MS was conducted at the Field Museum, with contributions by Nicola Sharratt, Laure Dussubieux, and Mark Golitko. Archaeological materials were exported from Peru under permit from the Ministry of Culture acuerdos 1659/792 on 18 December 2006 and 123 on 2 October 2003.

**Conflicts of Interest:** The authors declare no conflict of interest.

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
