# Peer review of "Archaeometric Approaches to Defining Sustainable Governance: Wari Brewing Traditions and the Building of Political Relationships in Ancient Peru"

_sustainability, doi:10.3390/su11082333_

Round 1
Reviewer 1 Report
This study linking residue analysis of Wari ceramics from Cerro Baúl to their form and composition is interesting, and represents in my opinion the direction in which compositional studies in archaeology should go in order to produce more analytically interesting results than simple sourcing studies.
That said, I do have two criticisms of this paper: 1) the data themselves could be more clearly presented in a single table (how many sherds analyzed, from where, with which techniques), and 2) I found the link to 'sustainability' through discussion of 'sustainable governance' to be a little forced.
With respect to (1), I infer from Fig.5 that only 3 sherds were analyzed for residues, but find that it's not actually clear from the text whether that is the case. If the sample is that small, surely it would be wise to recognize that the results are perhaps provisional. I also remain unsure how many experimental samples were analyzed, as it's not clear to me whether Table 1 is showing aggregate results or individual sherds.
With respect to (2), I think several questions are outstanding:
Why should we think that Cerro Baúl, or Wari generally, represents an instance of sustainable governance? It might - I'm not arguing with that per se - but the case needs to be made. For that matter, what makes one form of governance sustainable and another not? There's too much left unsaid/undefined here, in my opinion.
There is a brief mention in the final paragraph of these brewing practices persisting for four centuries, but the paper would be a lot more compelling if that were laid out explicitly, and at the beginning. i.e., the paper could be framed as:
"Cerro Baúl provides evidence that Wari colonial governance in Moquegua, maintained at least in part through ritual feasting centered around chicha consumption, persisted for four centuries. This implies that the practice, and the governmental system in which it was implicated, were remarkably sustainable, including during periods of more arid climate. This paper uses the results of ceramic sourcing and residue analyses of ceramics from the chicha brewery at Cerro Baul to explore the underpinnings of that sustainability."
A few specific comments:
p2
Line 68-69
The reference to the type of chicha preferred by the Wari should be substantiated with a citation, e.g., of Ref. 16.
p7
Line 211-214
"It is apparent that the brewery was a key consumer of the special ceramic workshop production represented by the Baúl chemical group. Temple and palace consumption contexts on the summit of Cerro Baúl adjacent to the brewery were also key consumers of the Baúl workshop ceramic wares."
I don't think that this is accurate - determining whether these areas were key consumers would depend on knowing the output and fate of the whole corpus of material produced by the workshop. Rather, it's the converse that we can say something about: the workshop was a key producer for the brewery.
p10-11
The discussion of the ingredients for chica de molle as likely more sustainable than those for chicha made from maize is interesting, and perhaps should come up in the introduction as well. I wonder if fuel use is also an issue, in terms of sustainability?
Author Response
Reviewer 1
Point 1) the data themselves could be more clearly presented in a single table (how many sherds analyzed, from where, with which techniques), and 2) I found the link to 'sustainability' through discussion of 'sustainable governance' to be a little forced.
With respect to (1), I infer from Fig.5 that only 3 sherds were analyzed for residues, but find that it's not actually clear from the text whether that is the case. If the sample is that small, surely it would be wise to recognize that the results are perhaps provisional. I also remain unsure how many experimental samples were analyzed, as it's not clear to me whether Table 1 is showing aggregate results or individual sherds.
Response 1: We have included a new table summarizing the corpus of ceramic sourcing data. ok
With respect to (2), I think several questions are outstanding: Why should we think that Cerro Baúl, or Wari generally, represents an instance of sustainable governance? It might - I'm not arguing with that per se - but the case needs to be made. For that matter, what makes one form of governance sustainable and another not? There's too much left unsaid/undefined here, in my opinion. There is a brief mention in the final paragraph of these brewing practices persisting for four centuries, but the paper would be a lot more compelling if that were laid out explicitly, and at the beginning. i.e., the paper could be framed as:
"Cerro Baúl provides evidence that Wari colonial governance in Moquegua, maintained at least in part through ritual feasting centered around chicha consumption, persisted for four centuries. This implies that the practice, and the governmental system in which it was implicated, were remarkably sustainable, including during periods of more arid climate. This paper uses the results of ceramic sourcing and residue analyses of ceramics from the chicha brewery at Cerro Baul to explore the underpinnings of that sustainability."
Response 2: We have also addressed the comments on sustainability by adding new data on fuel use and time depth per the reviewer’s comments. ok
Point 3: p2Line 68-69 The reference to the type of chicha preferred by the Wari should be substantiated with a citation, e.g., of Ref. 16.
Response 3: done.
Point 4: p7 Line 211-214 "It is apparent that the brewery was a key consumer of the special ceramic workshop production represented by the Baúl chemical group. Temple and palace consumption contexts on the summit of Cerro Baúl adjacent to the brewery were also key consumers of the Baúl workshop ceramic wares." I don't think that this is accurate - determining whether these areas were key consumers would depend on knowing the output and fate of the whole corpus of material produced by the workshop. Rather, it's the converse that we can say something about: the workshop was a key producer for the brewery.
Response 4: We believe the new table illustrates that the palace and brewery were key consumers of the Baul workshop’s production given the lack of that ceramic material elsewhere.
Point 5: p10-11 The discussion of the ingredients for chica de molle as likely more sustainable than those for chicha made from maize is interesting, and perhaps should come up in the introduction as well. I wonder if fuel use is also an issue, in terms of sustainability?
Response 5: We have also addressed the comments on sustainability by adding new data on fuel use and time depth per the reviewer’s comments.ok
Reviewer 2 Report
This paper is of great significance to apply cutting edge analytical technique to both experimental and archaeological ceramic samples that were presumably used for brewing and serving alcoholic beverages. Thus, the reviewer wishes this manuscript would be a little more improved in some points described below.
There must be an explanation of why and how this site was considered as a brewery.
What kind of contexts did they find large quantities of desiccated molle seeds? Was it indicative for brewing?
How the authors identified the decorated ceramics as brew-providing vessels? Is there any pictorial evidence which supports this assumption?
The aim and reason for choosing these methodologies must be clearly stated. The reviewer highly evaluates the use of DART-MS to identify the molecular record in archaeological remains so wishes authors to explain the merit of using this method compared to conventional organic residue analysis.
Experiments should be done on all the possible ingredients mentioned in the introduction, to strengthen the authors' point.
By the reviewer's point of view, terpin is a little too ubiquitous to identify a specific species. Need careful discription.
Would be better to provide mass spectra of archaeological findings to show they successfully identify these components.
Is there any potential contamination because of removing the surface of the archaeological ceramics? Plant-origin residues from soil can remain on the surface of pottery.
Author Response
Reviewer 2
Point 1: There must be an explanation of why and how this site was considered as a brewery. What kind of contexts did they find large quantities of desiccated molle seeds? Was it indicative for brewing? How the authors identified the decorated ceramics as brew-providing vessels? Is there any pictorial evidence which supports this assumption?
Response 1: We have elaborated on the description of the brewery to answer the reviewer’s first three comments and have discussed the identification of brewing vessels more formally.ok
Point 2: The aim and reason for choosing these methodologies must be clearly stated. The reviewer highly evaluates the use of DART-MS to identify the molecular record in archaeological remains so wishes authors to explain the merit of using this method compared to conventional organic residue analysis. Experiments should be done on all the possible ingredients mentioned in the introduction, to strengthen the authors' point. By the reviewer's point of view, terpin is a little too ubiquitous to identify a specific species. Need careful discription.
Response 2: We have strengthened the justification for the use of DART-MS and clarified that we tested all ingredients known to be used in chicha and present in the macro-botanical evidence from the brewery.
Point 3: Would be better to provide mass spectra of archaeological findings to show they successfully identify these components.
Response 3: We have added a figure showing spectra of archaeological ceramics. ok
Point 4: Is there any potential contamination because of removing the surface of the archaeological ceramics? Plant-origin residues from soil can remain on the surface of pottery.
Response 4: While potential contamination always requires careful treatment, we have elaborated our description on how we dealt with potential contamination issues in the residue analysis.
Reviewer 3 Report
Please see the comments attached.

Author Response
Point 1: Line 75 – elites were the only ones to drink chicha? If commoners did not have access then perhaps it could be worked into this sentence the overall restrictive nature of this ‘political beverage’.
Response 1: We have clarified the role of elites in chicha consumption and expanded on the description of the brewery as suggested.ok
Point 2: Line 87 – I think Figure 1 needs to be split into two figures. One is a map of Peru showing all of the sites mentioned in the text and the source locations of the clay discussed. And Cerro Baul is not even listed on the map. The other figure is a map of Cerro Baul. I see the word ‘brewery’ listed but it is not pointing to a building. As a result I do not know its location in relation to other building at the site. Perhaps an inset map of the brewery building would be a good idea.
Response 2: Figure 1 has been modified for greater clarity.
Point 3: Line 88 – I know the authors state in this sentence that if the reader wants to lern more about the brewery go read another article of theirs but I think it is useful here to include a couple of sentences outlining the brewery is – how identified/what indicates its function – because it seems to me that it isn’t a brewery until the chemical testing is done and that is reported here in this manuscript.
Response 3: We have expanded on the description of the brewery.ok
Point 4: Lines 106-114 – why is this paragraph italicized? I think a table listing the different number of sherds tested is useful here. This whole section needs clarification. The authors say that ‘another 20 samples from nearby consumption contexts were analyzed...’. What are these – middens? From which site are these 20 from? In Line 110, they talk about ’43 sherds from other summit contexts on Cerro baul...’. What other summit contexts? I need to know exactly where all of these sherds derive from. As it stands I have no idea. In Line 119 – “20 Wari samples from Baul – you mean Cerro baul – consumption contexts...”. What consumption contexts? And what do you mean by ‘consumption’? A table showing the sample number, frequency, context, and site – at a minimum – is warranted. I presume that all are from elite contexts, correct?
Response 4: The new table and changes to the description of the ceramic sourcing sample should clarify the sample as requested.
Point 5: Line 140 (figure 3) – In this figure there is all sorts of information presented for the first time. It is not stated Ceramic Paste Analysis Methods section that there is Wari-1Group, nor Mejia A-G groups. Both in the Ceramic Paste Analysis Methods section and in the caption for Figure 3 there needs to be a discussion about all of the groups presented.
Response 5: We have added further discussion of the groups present in the sample to clarify figure 3. ok
Line 144 – Wari capital of Ayacucho – not on map.
Line 152 – same issue as Figure 3 with how the data is presented.
Point 6: Line 159 – the discussion hre of the clay sources warrants a map listing all source loations.
Response 6: The clay source locations are published in Sharratt 2009.
Point 7: Line 164 – somewhere in here the authors need to discuss the issue of contamination and how they prevented it from getting into their experimental archaeology process and the archaeological ones both before and during their lab analysis.
Response 7: We have added a paragraph on contamination issues.
Line 166 – how long was the chicha left in the vessels? At least 10 days
Line 174 – Did Nash prepare this chicha de jora? If not, why not?yes
Response 8: We have expanded the description of the experimental data collection to answer these questions.
Line 218- is this the first time Cerro Mejia is mentioned? Is this site the one used in Figures 3 and 4?
Line 271 (Discussion) – It appears from the text that there were different people involved in different aspects of this. Potters produced the ceramic vessels and brewers produced the beer. I know this sounds obvious but want to make sure we are talking about specialists here.
Response 10: We have clarified this is the case.ok
Line 332 - Do you believe there were different chicha drinks for different feasting events? And, can you make it clear if the chicha you discuss throughout this manuscript is made with pepperberry?
Response 11: We believe chicha de jora and de molle were made in the region by different groups at different times.
Overall, I think the bibliography is extensive and relevant. I think new maps need to be added to the text. For the most part, my comments are minor. I do think this is an important contribution because it presents INAA data along with the chemical data. So few breweries have ever been found archaeologically around the world that I believe it is necessary to include a few sentences about this discovery. For issues of contamination it is important for the authors to address how they excavated the sherds and whether they came in contact with anything that might cloud their results. Similarly, what was the protocol in the lab for testing the sherds. There is no real discussion how the samples were taken. My focus on contamination with this manuscript is really about being able to replicate the methods at another lab elsewhere in the country. I am very pleased with such a study being published because it advances our knowledge on so many different topics, from the types of analyses performed to cultural questions pertaining to production strategies involving ceramics and alcoholic beverages and political identity and economy.
Response 12: We appreciate the reviewer’s comments and have addressed the issues summarized per our above responses.